# Digital Health Service for Identification of Frailty Risk Factors in Community-Dwelling Older Adults: The SUNFRAIL+ Study Protocol

**DOI:** 10.3390/ijerph20053861

**Published:** 2023-02-21

**Authors:** Vincenzo De Luca, Grazia Daniela Femminella, Lisa Leonardini, Lola Patumi, Ernesto Palummeri, Isabella Roba, Walter Aronni, Stefano Toccoli, Simona Sforzin, Fortunata Denisi, Anna Maddalena Basso, Manuela Ruatta, Paola Obbia, Alessio Rizzo, Moira Borgioli, Claudio Eccher, Riccardo Farina, Diego Conforti, Lorenzo Mercurio, Elena Salvatore, Maurizio Gentile, Marialuisa Bocchino, Alessandro Sanduzzi Zamparelli, Giulio Viceconte, Ivan Gentile, Carlo Ruosi, Nicola Ferrara, Gabriella Fabbrocini, Annamaria Colao, Maria Triassi, Guido Iaccarino, Giuseppe Liotta, Maddalena Illario

**Affiliations:** 1Dipartimento di Sanità Pubblica, Università Degli Studi di Napoli Federico II, 80131 Napoli, Italy; 2Dipartimento di Scienze Mediche Traslazionali, Università Degli Studi di Napoli Federico II, 80131 Napoli, Italy; 3Programma Mattone Internazionale Salute, Azienda ULSS 4 Veneto Orientale, 30027 San Donà di Piave, Italy; 4A.li.sa. (Azienda Ligure Sanitaria), Regione Liguria, 16121 Genova, Italy; 5Dipartimento di Cure Primarie e Attività Distrettuali, Azienda Sociosanitaria Ligure 4, 16043 Chiavari, Italy; 6Dipartimento Cure Primarie, Azienda Provinciale per i Servizi Sanitari di Trento, 38123 Trento, Italy; 7Cooperativa Sociale “Res Omnia”, 89128 Reggio Calabria, Italy; 8Direzione Professioni Sanitarie, Azienda Sanitaria Locale Cuneo 1, 12100 Cuneo, Italy; 9Rete Della Cronicità e Fragilità, Azienda Sanitaria Locale Cuneo 1, 12100 Cuneo, Italy; 10Dipartimento di Scienze Della Sanità Pubblica e Pediatriche, Università Degli Studi di Torino, 10126 Torino, Italy; 11Settore Sistemi Organizzativi e Risorse Umane Della Direzione Sanità e Welfare, Regione Piemonte, 10144 Torino, Italy; 12Unità Operativa Complessa Progettazione, Sviluppo, Formazione e Ricerca, Azienda Unità Sanitaria Locale Nord-Ovest, 56121 Pisa, Italy; 13eHealth Unit, Fondazione Bruno Kessler, 38123 Trento, Italy; 14Dipartimento Salute e Politiche Sociali, Provincia Autonoma di Trento, 38122 Trento, Italy; 15Dipartimento di Scienze Biomediche Avanzate, Università Degli Studi di Napoli Federico II, 80131 Napoli, Italy; 16Dipartimento di Medicina Clinica e Chirurgia, Università Degli Studi di Napoli Federico II, 80131 Napoli, Italy; 17Dipartimento di Biomedicina e Prevenzione, Università Degli Studi di Roma Tor Vergata, 00133 Roma, Italy

**Keywords:** frailty, active and healthy aging, pre-frailty, bio–psycho–social domains, frailty risk factors, community-dwelling older adults, nursing, study protocol

## Abstract

This article reports the study protocol of a nationwide multicentric study in seven Italian regions aimed at assessing the effectiveness of a digitally supported approach for the early screening of frailty risk factors in community-dwelling older adults. SUNFRAIL+ is a prospective observational cohort study aimed at carrying out a multidimensional assessment of community-dwelling older adults through an IT platform, which allows to connect the items of the SUNFRAIL frailty assessment tool with a cascading multidimensional in-depth assessment of the bio–psycho–social domains of frailty. Seven centers in seven Italian regions will administer the SUNFRAIL questionnaire to 100 older adults. According to the answers provided by older adults, they will be subjected to one or more validated in-depth scale tests in order to perform further diagnostic or dimensional evaluations. The study aims to contribute to the implementation and validation of a multiprofessional and multistakeholder service model for the screening of frailty in community-dwelling older adult population.

## 1. Introduction

Demographic changes in Europe and the progressive aging of the population are likely to affect the planning and delivery of health and social services due to the increasing demand for care services and the complexity of the health needs of the over 65 population [1]. Older adults with multimorbidity are a vulnerable subgroup of the population that has been found to have a higher mortality from COVID-19 and a higher risk of adverse outcomes [2]. Furthermore, the COVID-19 pandemic has highlighted the deterioration of the quality of life of older adults due to the drastic change in their social habits and reduced access to health and social services [3].

Frailty in older adults results in reduced physiological reserve, which leads to an increased risk of falls, hospitalization, disability and death [4]. The determinants of frailty syndrome are multiple and interconnected. In addition to the natural physiological changes of aging, multimorbidities, malnutrition, living environment, genetics, social relations and lifestyle also influence the onset of frailty [5]. Frailty limits regular physical and social activity and their related health benefits for older people [6]. The increasing prevalence of frailty has different implications in terms of health needs that require different interventions depending on the time of onset and the socioeconomic background of individuals [7]. Frail people represent an increasing economic burden on healthcare systems [8]. Early screening for frailty risk in community-dwelling older people enables preventive intervention on the clinical and social determinants of frailty and, thus, the prevention of adverse events [9].

The centrality of nursing services in the promotion of health in the older adult population through prevention, education and health information is certainly a topic that has been discussed and added to the agenda of public health systems in recent years [10]. Many Italian regions and all actors involved in the working groups activated by the Programma Mattone Internazionale Salute (ProMIS), the Italian Ministry of Health program for the internationalization of regional health systems [11], have been involved in various ways in the implementation of the family and community nurse (FCN) professional profile. Italian regions are progressively introducing FCN into their health system in a pilot form or through dedicated policies, defining its boundaries, actions and relations [12,13]. The FCN is pivotal for meeting the twin challenges of an aging population and the increasing incidence of long-term conditions in order to prevent adverse events and enable health systems to encourage and ensure the improvement of health and well-being instead of focusing exclusively on the treatment of illness [14].

The European project SUNFRAIL proposes an innovative and integrated approach that takes into account all of the various factors that influence an individual’s state of health towards frailty: environmental, medical, educational, economic and psychological. The SUNFRAIL approach starts from the assumption that frailty is a reversible condition if detected at an early stage and that it must, therefore, be identified first and foremost in primary care and community settings in order to prevent dependency and disability [15].

As a result, the SUNFRAIL project developed a tool for the early identification of frailty in the over 65 population that can be used in different contexts, consisting of only nine items, which can be linked to other in-depth tools for the assessment of specific frailty domains [16]. The SUNFRAIL tool aims to detect frailty risk factors at an early stage, guiding subsequent diagnostic assessments for the identification of health promotion and disease prevention targeted interventions (Figure 1).

The SUNFRAIL tool enables very early screening in primary care, community and specialist services, guiding them to subsequent in-depth clinical and nonclinical investigations with respect to the identified domains of frailty (i.e., physical, cognitive, nutritional and social). It represents a synthesis tool for socio-health integration services, where the physical dimension is not exhaustive but must be associated with the biological, psychological and socio-relational dimensions. Primary care for older adults with multimorbidity is characterized by the fragmentation of care services (general practitioners, geriatricians, territorial specialists, rehabilitators, FCNs, etc.). In this sense, information technology (IT) represents a very powerful tool to support prevention and healthcare activities. In fact, IT has become increasingly widespread and available in recent years, although it is currently only partially validated [17].

Therefore, the ProMIS Working Group on Frailty designed a new service model for the frailty screening of community-dwelling older adults by means of the SUNFRAIL+ platform, an IT tool that supports healthcare professionals by linking the items of the SUNFRAIL tool to additional scales aimed at assessing impaired domains and enabling him/her to develop appropriate intervention strategies. This IT-supported multidimensional approach improves the availability of data on the target population for health professionals to develop targeted interventions in the domains identified as being at risk. In this model, the FCN can act as the first point of contact for individuals and families and play a leading role in disease prevention and health promotion, turning to specialists when more expertise is needed.

## 2. Materials and Methods

### 2.1. Study Design

The SUNFRAIL+ study protocol was developed through a prospective observational cohort study. The study will carry out a multidimensional assessment of older adults, through the SUNFRAIL+ platform, which allows the linking of each item of the SUNFRAIL tool with a cascading multidimensional in-depth assessment of the bio–psycho–social domains of frailty [18]. The study aims to assess the impact of early, integrated and simultaneous “taking care” by health systems on the quality of life of community-dwelling older adults through the SUNFRAIL+ platform. In addition, the study aims to verify the effect of SUNFRAIL+ on the appropriateness of care by the healthcare system and the impact of IT solutions on the organization of services and the workload of professionals.

### 2.2. Centers

It is a multicentric study, promoted by ProMIS, which includes the following centers and regions (Figure 2):Department of Public Health of the University of Naples Federico II (Campania Region), as coordinator;Azienda Sociosanitaria Ligure 4 (Liguria Region);Azienda Provinciale per i Servizi Sanitari (Autonomous Province of Trento);“Res Omnia” Social Cooperative (Calabria Region);Azienda Sanitaria Locale Cuneo 1 (Piedmont Region);Azienda Unità Sanitaria Locale Nord-Ovest (Tuscany Region);Department of Biomedicine and Prevention of the University of Rome Tor Vergata (Lazio Region).

### 2.3. Participants

Each center will independently recruit 100 ambisexuals, randomly selected older adults. Subjects who meet the following inclusion criteria are eligible to participate: age ≥65 years; living at home; accessing participating centers for social and healthcare services other than those covered by this study; and ability to sign the informed consent. Subjects will be excluded if they are <65 years old; residents in care facilities (hospital and nursing home); have overt frailty or disability; are already enrolled in home care; and are unable to understand the study aims and sign the informed consent.

### 2.4. Intervention

The study will have a duration of 14 months: 6 months for the first multidimensional assessment (enrolment), 6 months “taking charge” (observation) and 2 months of data analysis and evaluation (follow-up) (Figure 3).

Each center will administer the SUNFRAIL tool to the enrolled older adults and, according to the specific frailty risk factors identified by the answers provided, they will be subjected (at the same time or subsequently) to one or more validated scales (already included in the SUNFRAIL+ platform) in order to perform further diagnostic or dimensional evaluations in the domains identified as being at risk.

In addition, demographic, anamnestic and anthropometric variables will be acquired. The multidimensional assessment through the platform will be carried out by trained personnel.

During the observation phase, older adults will be offered different health promotion activities, depending on the multidimensional assessment. Healthy older adults will be offered collective physical exercise programs and nutritional recommendations. In addition, services will be provided to promote socialization and educational modules dedicated to fall prevention. For subjects with chronic diseases, in addition to health promotion, services for patient empowerment and improved adherence to treatment goals will be provided.

During the follow-up phase, 6 months after enrolment, the SUNFRAIL tool and in-depth tests will be administered again. In addition, during the follow-up, the extent to which the exposure to the interventions has influenced the quality of life of older adults will be measured, and the number of users enrolled by the territorial services based on user care needs will be documented. In addition, the level of satisfaction and usability of the SUNFRAIL+ solution by health and social care professionals will be measured.

### 2.5. Outcomes

The data from the study will be collected through the following instruments and self-assessment questionnaires:Frailty multidimensional screening: SUNFRAIL tool [16];Prescription adherence: medication adherence report scale (MARS) [19];Nutrition: assessment of adherence to the Mediterranean diet (PREDIMED) [20] and mini-nutritional assessment (MNA) [21];Physical activity: short physical performance battery (SPPB) [22];Fall risk: age-friendly environment assessment tool (AFEAT) [23] and time up and go test [24];Cognitive decline: quick mild cognitive impairment (QMCI) [25] and general practitioner assessment of cognition (GPCOG) [26];Loneliness: geriatric depression scale (GDS) [27];Support network: social provisions scale (SPS) [28];Socioeconomic conditions: self-assessment questionnaire (MUSE) [29];Quality of life: SF-12 health survey [30];Usability: system usability scale [31].

The adherence to medical visits will be measured by means of a questionnaire in which older adults will be asked how many times in the last 3 months they visited the GP, received home nursing visits, went to hospital, went to the emergency room, and had specialist examinations, blood tests or other diagnostic examinations. In addition, older adults will be asked whether they have difficulties in making an appointment with the GP or in getting to the GP’s clinic.

### 2.6. Statistical Analysis

All complete questionnaires will be analyzed. The statistical analysis will be:(1)Descriptive and exploratory (ADE) with respect to the study population, bringing to light any “features” present in the data;(2)Confirmatory (ADC), focusing on confirming or falsifying the appropriateness of early, integrated and simultaneous care.

For the descriptive analysis, the absolute and relative frequency distributions of all qualitative variables will be calculated, while for the quantitative or semiquantitative variables (e.g., scales from 1 to 10), the main summary and variability indices (mean, median, minimum, maximum, standard deviation, range and interquartile difference) will be calculated.

The main statistical tests will be used for confirmatory analyses, depending on the types of variables: chi-square test, McNemar test for dichotomous data, parametric t-test and Wilcoxon’s nonparametric test for paired data (before–after). All tests will be two-sided, with a 5% significance level. The multiple linear regression model will be used to examine the contribution of the parameters measured during the study (explanatory variables) on the quality of life (score measured by the SF-12 questionnaire, dependent variable).

The analyses will be performed using IBM SPSS Statistics software for Windows, version 26, Armonk, NY, USA, IBM Corp.

With a type 1 error of 5% and a study power of 80%, accepting a 95% confidence level and targeting an improvement of 10% in the SF12 score, the requested sample size will be composed of 195 subjects. The number of 700 clients exceeds the needed number of participants, which can be useful to perform nested analysis.

### 2.7. Ethics Statement

Each center is responsible for conducting the study in accordance with good clinical practice and international quality standards for the design, conduct and dissemination of studies involving human subjects [32]. The study protocol, informed consent and any accompanying material provided to the patient have been approved by the local Ethics Committee of Università “Federico II”–Azienda Ospedaliera di Rilievo Nazionale “A. Cardarelli” (record number: 284/22). The research protocol was registered at ClinicalTrials.gov on 9 December 2022 (registration number: NCT05646472).

## 3. Discussion

The study aims to contribute to the implementation and validation of a multiprofessional and multistakeholder service model for the screening of frailty in community-dwelling older adult population, using the SUNFRAIL+ platform, an IT tool that supports the health and/or social care professional by linking the elements of the SUNFRAIL tool to further scales aimed at assessing impaired bio-psycho-social frailty domains and enabling the development of integrated intervention strategies. The study addresses the fragmentation of care services for older adults (general practitioners, geriatricians, territorial specialists, rehabilitators, FCN, etc.). SUNFRAIL+ responds to the need to support health professionals in the implementation of integrated health and care services through multidisciplinary pathways at home or in common aggregation points. The approach responds to the consolidated need for the early detection of risk conditions that lead to the progressive loss of autonomy in order to reduce the impact of health conditions that require a strong commitment of socioeconomic resources to ensure adequate care and cure that comprise the quality of a person’s existence. The study not only concerns the FCN but aspires to provide a model integrating different services of the health ecosystem to maximize health and well-being, empowering individuals, families and communities, motivating them to take responsibility for their own health. Through the use of Sunfrail+, social and healthcare professionals are supported and motivated to encourage and/or activate care interventions, mainly aimed at preventing falls and cognitive decline and combating social and psychological distress.

SUNFRAIL+ will improve the availability of data on the target population to health and social care professionals, enabling them to develop targeted prevention and health promotion interventions in the domains identified as being at risk, using the resources of the community in which the older adult lives [33].

The availability of older people health data will allow an informed and integrated intervention between local social and health services and the community, with the aim of improving the independent life of older adults, helping them to play an active role in society [34]. The study will have an expected impact on the quality of life of older adults and the improvement of health outcomes in the risk factors of frailty, deriving from participation in collective and individual prevention and health promotion services, such as adapted physical activity programs, primary and secondary nutritional interventions, fall prevention, adherence to drug therapies and medical visits, socialization and psychological support.

The SUNFRAIL+ IT platform makes the pathway more structured and facilitates data collection, especially among citizens in areas with limited access to healthcare services. The availability of an increasing volume of health data on the older adult population will enable healthcare systems to move from reactive disease management to health promotion. The adoption of digital solutions is crucial for the continuous improvement of diagnosis, treatment and personalization of care [35,36]. Frailty screening by means of the SUNFRAIL+ tool will enable the activation of any subsequent diagnostic tests at the relevant healthcare facilities.

### Strengths and Limitations

One of the strengths of the study is the participation of several centers, providing different services but all interacting with the target population along the SUNFRAIL specific domains. This will make it possible to demonstrate that the identification of risk factors for frailty in older adults and their enrolment in prevention and health promotion services should not only take place at a dedicated healthcare facility but in different settings and at different levels of care.

The main limitation of the study is that the intervention strategies of the health systems involved and the services available from health organizations are not uniform. This does not allow the study to be able to investigate the effectiveness of prevention and health promotion interventions to be linked to the screening of frailty risk factors. The exploitation of the study outcomes depends very much on the ability of the centers to link the screening of frailty’s risk factors with the services currently available in the participating centers for prevention and health promotion.

## 4. Conclusions

The SUNFRAIL+ nationwide multicenter observational study aims to demonstrate the effectiveness of a digitally supported approach for the early screening of frailty risk factors in community-dwelling older adults. The study results will encourage the implementation and validation of a multiprofessional and multistakeholder service model for frailty prevention in community-dwelling older adult populations.

## Figures and Tables

**Figure 1 ijerph-20-03861-f001:**
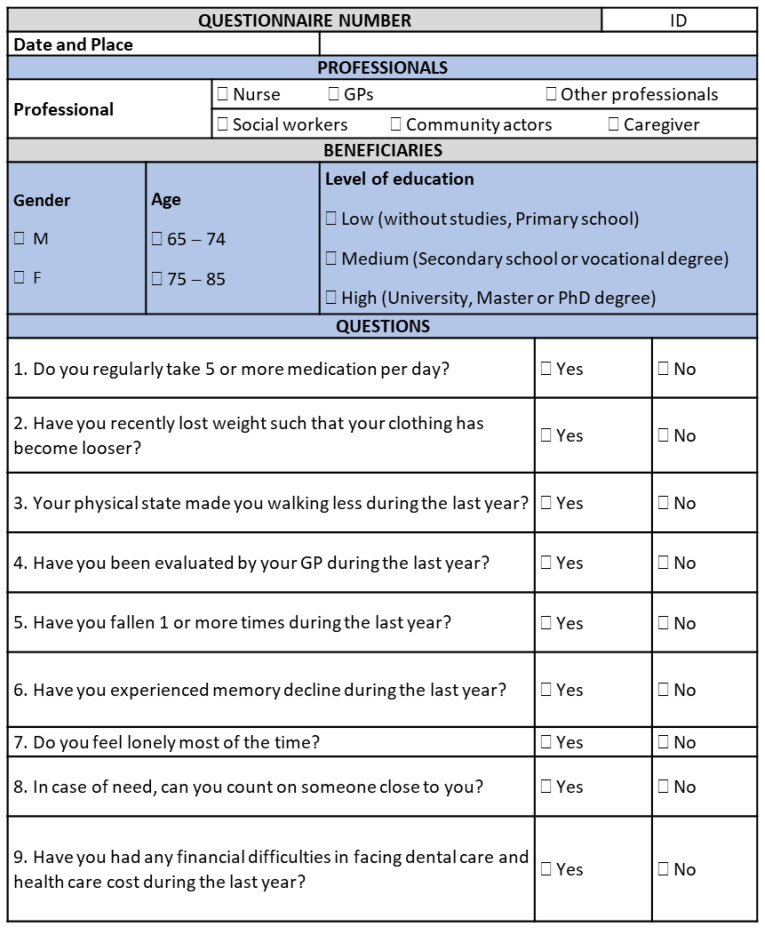
The SUNFRAIL assessment tool.

**Figure 2 ijerph-20-03861-f002:**
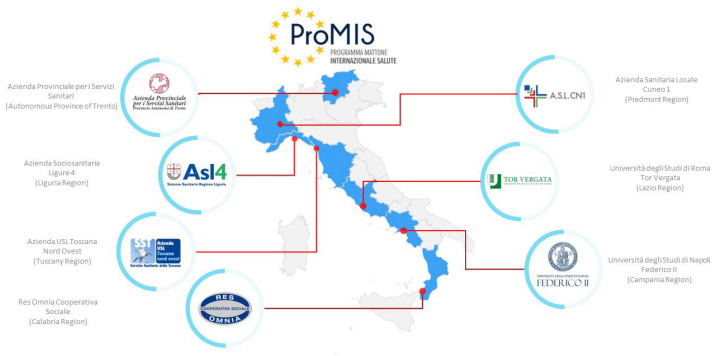
Location of the centers participating in SUNFRAIL+ on the map representing the Italian regions (Italy).

**Figure 3 ijerph-20-03861-f003:**
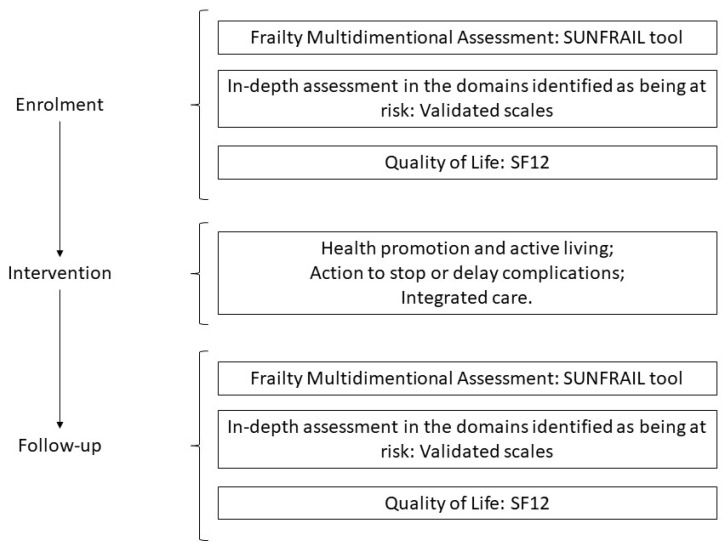
SUNFRAIL+ study design and planned intervention.

## Data Availability

The data of present study will be available in an anonymized form upon documented instruction of the data controllers (Centres) to the responsible for data processing (Fondazione Bruno Kessler). The data will not be publicly available due to ethical and privacy reasons.

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
