# Peer review of "Digital Health Service for Identification of Frailty Risk Factors in Community-Dwelling Older Adults: The SUNFRAIL+ Study Protocol"

_ijerph, 2023, doi:10.3390/ijerph20053861_

Round 1

Reviewer 1 Report

Abstract

The abstract is a little wordy and can be simplified.

Introduction 

Well written but some citations are missing. Please recheck citations and vocabulary.

Methods

consider including the core elements of the intervention, beyond the questionnaire that shows description of measure and expected outcome.

DISCUSSION

A bit short. Perhaps expand and discuss hopes and expectations for outcomes and future implications

LIMITATIONS AND STRENGTHS

It seems a little odd and abrupt to end with this. I feel like we are left hanging and the paper is incomplete. Consider restructuring.

FIGURE 2

Very difficult to read and understand.

Author Response

Authors thank you for giving the opportunity to submit a revised draft of the manuscript. We appreciate the time and effort that you dedicated to providing feedback on our manuscript and are grateful for the insightful comments on and valuable improvements to our paper. We have incorporated most of the suggestions made by you. Please see below for a point-by-point response to your comments and concerns

Abstract

The abstract is a little wordy and can be simplified.

As suggested by the reviewer, the authors simplified the abstract.

Introduction

Well written but some citations are missing. Please recheck citations and vocabulary.

As suggested by the reviewer, the authors included the missing citations.

Methods

consider including the core elements of the intervention, beyond the questionnaire that shows description of measure and expected outcome.

As suggested by the reviewer, the authors included the core elements of the intervention (lines 173-178).

DISCUSSION

A bit short. Perhaps expand and discuss hopes and expectations for outcomes and future implications.

As suggested by the reviewer, the authors extended paragraph 4 with expected outcomes and future implications.

LIMITATIONS AND STRENGTHS

It seems a little odd and abrupt to end with this. I feel like we are left hanging and the paper is incomplete. Consider restructuring.

As suggested by the reviewer, the authors included a conclusion paragraph.

FIGURE 2

Very difficult to read and understand.

As suggested by the reviewer, the authors revised the legend in Figure 2 and modified some graphic aspects to make it more understandable.

Reviewer 2 Report

The paper titled "Digital health service for identification of frailty risk factors in community-dwelling older adults: the SUNFRAIL+ study protocol" is a well written and detailed report. This is a very useful study that will be of benefit in the context of global aging population. Kindly see my comments as below;

1. The paper would benefit from a flowchart that depicts the study design and intervention planned. 

2. There is a lack of clarity regarding the basis on which specific alerts will be triggered and what are the in depth assessments that will be conducted following this. Kindly provide sufficient details regarding this aspect.

3. Details regarding the health promotion activities (lines 167-169) planned during the observation phase are lacking. This aspect may require further elaboration.

4. It will be optimal if there is more clarity on the validated tools that are planned to be used during the follow up phase. (lines 170-175)

5. The discussion lacks sufficient depth. Perhaps the authors can justify the need and significance of a digital health service such as SUNFRAIL+ in the prevention of frailty and how this can impact at an individual, community as well as regional level.

6. The paper would benefit from a short and crisp conclusion. 

Author Response

Authors thank you for giving the opportunity to submit a revised draft of the manuscript. We appreciate the time and effort that you dedicated to providing feedback on our manuscript and are grateful for the insightful comments on and valuable improvements to our paper. We have incorporated most of the suggestions made by you. Please see below for a point-by-point response to your comments and concerns

The paper titled "Digital health service for identification of frailty risk factors in community-dwelling older adults: the SUNFRAIL+ study protocol" is a well written and detailed report. This is a very useful study that will be of benefit in the context of global aging population. Kindly see my comments as below;

  1. The paper would benefit from a flowchart that depicts the study design and intervention planned.

As suggested by the reviewer, the authors included Figure 3 to depicts the study design and intervention planned

  1. There is a lack of clarity regarding the basis on which specific alerts will be triggered and what are the in depth assessments that will be conducted following this. Kindly provide sufficient details regarding this aspect.

As suggested by the reviewer, the authors clarified how in-depth assessments will be conducted (lines 95-98 and lines 164-168). The study involves a multidimensional assessment through the SUNFRAIL tool, for the identification of risk factors for frailty in the nine domains. Based on the responses of the older adults, validated scales will be administered for assessment the domains identified as being at risk.

  1. Details regarding the health promotion activities (lines 167-169) planned during the observation phase are lacking. This aspect may require further elaboration.

As suggested by the reviewer, the authors included the core elements of the intervention (lines 173-178).

  1. It will be optimal if there is more clarity on the validated tools that are planned to be used during the follow up phase. (lines 170-175)

The validated scales used for enrolment and follow-up are those listed in section 2.5.

  1. The discussion lacks sufficient depth. Perhaps the authors can justify the need and significance of a digital health service such as SUNFRAIL+ in the prevention of frailty and how this can impact at an individual, community as well as regional level.

As suggested by the reviewer, the authors extended paragraph 4 with expected outcomes and future implications.

  1. The paper would benefit from a short and crisp conclusion.

As suggested by the reviewer, the authors included a conclusion paragraph.

Reviewer 3 Report

Dear Authors, 

Thank you for your work in this very important topic. Your study is well designed and your paper is well written. My only suggestions are:

1. To provide the assumptions that were used to determine your sample size.

2. Be more specific about how your statistical analysis will answer your research questions. You mention outcomes and statistical tests, but you should connect them. 

3. Explain the process of validating the results of the tool for early detection of frailty.

Author Response

Authors thank you for giving the opportunity to submit a revised draft of the manuscript. We appreciate the time and effort that you dedicated to providing feedback on our manuscript and are grateful for the insightful comments on and valuable improvements to our paper. We have incorporated most of the suggestions made by you. Please see below for a point-by-point response to your comments and concerns

Dear Authors,

Thank you for your work in this very important topic. Your study is well designed and your paper is well written. My only suggestions are:

  1. To provide the assumptions that were used to determine your sample size.

As suggested by the reviewer, the authors included in paragraph 2.6 the assumptions that were used to determine the sample size.

  1. Be more specific about how your statistical analysis will answer your research questions. You mention outcomes and statistical tests, but you should connect them.

Thank you for suggestions. The main outcome of the study is the SF12 score to measure quality of life before and after the intervention. This will be assessed with the T-Test for paired sample.

  1. Explain the process of validating the results of the tool for early detection of frailty.

Thank you for suggestions. The hypothesis of the study is that the SUNFRAIL+ methodology is able to improve the quality of life of the beneficiaries, so that the validation of the methodology should be expressed by the improvement of quality of life. In fact, usually quality of life decrease over time in older adults

Reviewer 4 Report

Lines 63-64: Please qualify "prevalence" by making it "increasing prevalence" because prevalence can either increase or decrease.

Figure 1: In Q1 (the first item), what is the reason behind using 5? Does the use of less than 5 medications prevent against frailty? The items in the questionnaire/assessment tool are scanty and there are not enough items about the risk factors of frailty as indicated in previous studies.

Lines 122-124: The tool does not have enough items/questions that can be linked to multidimensional in-depth assessment. I suggest that you add more relevant questions to the tool.

Line 147: The study is supposed to be a prospective cohort study, however, the authors only mentioned one group of participants. How about the control/non-exposed group? Also, the recruitment of 100 adults were mentioned, how would you address loss to follow up which is always a big problem in cohort studies?

Line 167: The authors mentioned that the study would be a prospective study which is a type of observational study but there would be an exposure to interventions (line172). This sounds more like an intervention/experimental study rather than an observational study. This is confusing.

Lines 195-197: Asking the study participants about the number of times they visited the GP , received home nursing visits etc will lead to recall bias. The participants are more likely to give you answers that are not true.

Also, doing in-depth tests might be too tiring for the older adults. I will suggest that minimal tests should be conducted especially the ones that are not invasive.

Author Response

Authors thank you for giving the opportunity to submit a revised draft of the manuscript. We appreciate the time and effort that you dedicated to providing feedback on our manuscript and are grateful for the insightful comments on and valuable improvements to our paper. Please see below for a point-by-point response to your comments and concerns.

Lines 63-64: Please qualify "prevalence" by making it "increasing prevalence" because prevalence can either increase or decrease.

As suggested by the reviewer, the authors revised the sentence.

Figure 1: In Q1 (the first item), what is the reason behind using 5? Does the use of less than 5 medications prevent against frailty? The items in the questionnaire/assessment tool are scanty and there are not enough items about the risk factors of frailty as indicated in previous studies.

Polypharmacy is a defined risk for frailty. Five or more medicines were used by Gnjidic D., et al. to identify community-dwelling older men at risk of different adverse outcomes.

10.1016/j.jclinepi.2012.02.018

Lines 122-124: The tool does not have enough items/questions that can be linked to multidimensional in-depth assessment. I suggest that you add more relevant questions to the tool.

The study involves a multidimensional assessment through the SUNFRAIL tool, for the identification of risk factors for frailty in the nine domains (polypharmacy, nutrition, physical activity, fall, adherence to medical visits, cognitive decline, loneliness, social support, and economic constraints). Based on the responses of the older adults, validated scales will be administered for assessment of each domain identified as being at risk.

Line 147: The study is supposed to be a prospective cohort study, however, the authors only mentioned one group of participants. How about the control/non-exposed group? Also, the recruitment of 100 adults were mentioned, how would you address loss to follow up which is always a big problem in cohort studies?

It is a prospective observational cohort study. It is NOT a case-control study.

Line 167: The authors mentioned that the study would be a prospective study which is a type of observational study but there would be an exposure to interventions (line172). This sounds more like an intervention/experimental study rather than an observational study. This is confusing.

This is an observational study because the intervention does not change clinical practice. They will be subjected to the regular treatment the older adults would receive from the healthcare organisations involved.

Lines 195-197: Asking the study participants about the number of times they visited the GP, received home nursing visits etc will lead to recall bias. The participants are more likely to give you answers that are not true.

The questionnaire on Adherence to Medical visits helps us to understand whether older adult is regularly monitored by healthcare personnel, and whether they have had an acute event of which his GP is unaware.

Also, doing in-depth tests might be too tiring for the older adults. I will suggest that minimal tests should be conducted especially the ones that are not invasive.

Thank you for your suggestions. Older adults will be subjected, at the same time or subsequently, to validated scales, only in the domains identified as being at risk.

Round 2

Reviewer 4 Report

The paper should be accepted in the present form.